# Hybrid Mamba for Few-Shot Segmentation

**Qianxiong Xu**[1], **Xuanyi Liu**[2], **Lanyun Zhu**[3], **Guosheng Lin**[1]\*, **Cheng Long**[1]\*, **Ziyue Li**[4], **Rui Zhao**[5]

[1]S-Lab, Nanyang Technological University    [2]Peking University
[3]Singapore University of Technology and Design    [4]University of Cologne    [5]SenseTime Research
`{qianxiong.xu, gslin, c.long}@ntu.edu.sg, xuanyi@stu.pku.edu.cn,`
`lanyun_zhu@mymail.sutd.edu.sg, zlibn@wiso.uni-koeln.de, zhaorui@sensetime.com`

## Abstract

Many few-shot segmentation (FSS) methods use cross attention to fuse support foreground (FG) into query features, regardless of the quadratic complexity. A recent advance Mamba can also well capture intra-sequence dependencies, yet the complexity is only linear. Hence, we aim to devise a cross (attention-like) Mamba to capture inter-sequence dependencies for FSS. A simple idea is to scan on support features to selectively compress them into the hidden state, which is then used as the initial hidden state to sequentially scan query features. Nevertheless, it suffers from (1) support forgetting issue: query features will also gradually be compressed when scanning on them, so the support features in hidden state keep reducing, and many query pixels cannot fuse sufficient support features; (2) intra-class gap issue: query FG is essentially more similar to itself rather than to support FG, i.e., query may prefer not to fuse support features but their own ones from the hidden state, yet the success of FSS relies on the effective use of support information. To tackle them, we design a hybrid Mamba network (HMNet), including (1) a support recapped Mamba to periodically recap the support features when scanning query, so the hidden state can always contain rich support information; (2) a query intercepted Mamba to forbid the mutual interactions among query pixels, and encourage them to fuse more support features from the hidden state. Consequently, the support information is better utilized, leading to better performance. Extensive experiments have been conducted on two public benchmarks, showing the superiority of HMNet. The code is available at `https://github.com/Sam1224/HMNet`.

## 1 Introduction

In the realm of computer vision, the advent of deep learning has ushered in remarkable advancements, e.g., in semantic segmentation [14, 26, 33, 62, 63, 65, 67]. However, the realization of such feats demands extensive time and human efforts dedicated to annotating pixel-wise masks. Furthermore, semantic segmentation methods will falter when confronted with previously unseen classes, thus impeding the generalization of segmentation to arbitrary classes. Inspired by the phenomenon that human can learn to recognize new objects by referring to a handful of samples, researchers have introduced few-shot segmentation (FSS) [34, 36, 46, 59, 66], which aims to use a few manually annotated support samples to help the segmentation of a query image, with arbitrary classes.

Recent advances in FSS consist of prototypical methods [2, 40, 46, 59] and attention-based methods [31, 48, 54, 60]. According to the support annotations, prototypical methods extract support foreground (FG) and compress their features into single or a few prototypes, which are then used to segment the query image through either feature comparison [2, 46] or feature fusion [40, 59].

---

\*Co-corresponding authors

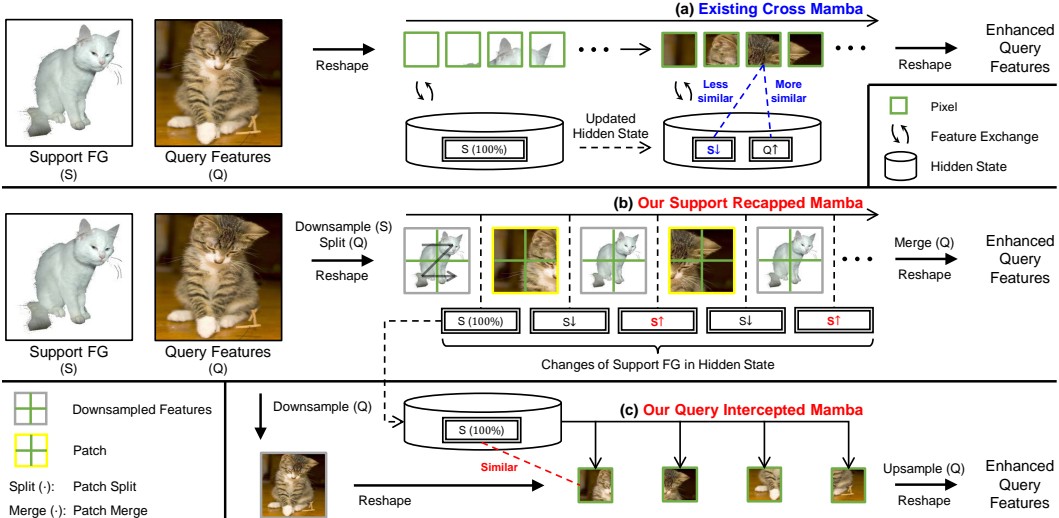

Figure 1: Illustrations of (a) existing cross Mamba, (b) our support recapped Mamba (SRM); and (c) our query intercepted Mamba (QIM). **In (a)**, the support features are firstly scanned and selectively compressed into the hidden state, which is expected to be fused into query FG. Nevertheless, (1) with the scan on query, the compressed support FG is gradually reduced, and (2) query FG is essentially more similar to itself rather than support FG. Thus, the support FG cannot well enhance the query FG features. **In (b) and (c)**, we design (1) a SRM to periodically re-scan the support FG, so the hidden state always contain sufficient support features, and (2) a QIM to intercept the mutual interactions among query pixels, thus, they are forcibly fused with support features.

Unfortunately, the compression would inevitably lose much support FG information [47] and disrupt the structure of FG objects [52]. Instead, others employ cross attention [41] to fuse query features with the uncompressed support FG features, so as to prevent from the information loss. Nevertheless, the computational complexity of attention is quadratic to the pixel number, hindering the processing of larger images and the deployment of more attention blocks.

Recently, selective state space model (SSM) [5, 6], also known as Mamba, appears to be an alternative approach to attention, which can effectively capture long-range dependencies but only with linear complexity, and has shown promising results in various vision tasks [25]. Specifically, Mamba flattens the 2D image features into a 1D pixel sequence, and iteratively scans the sequence to (1) selectively compress each pixel's features into a finite hidden state, and (2) use the hidden state, including the filtered features of previously scanned pixels, to enhance the current pixel's features. Although there exist many follow-up works of Mamba, they mainly focus on capturing the dependencies within the same image (i.e., self Mamba) [25, 68], and not much attention has been paid to the case where the dependencies between different images should be captured (i.e., cross Mamba), which is required by FSS. Therefore, we aim to design a cross Mamba for FSS in this paper.

As shown in Figure 1(a), one possible solution [32, 61] is to put the query pixel sequence right after the support FG pixel sequence, and use the standard Mamba to scan the concatenated sequence. Once the scan on support is finished, the hidden state has already included useful support FG features, which can then enhance the query features (i.e., cross attention) during the scan on query. Nevertheless, two issues would arise: (1) *Support forgetting*: With the scan on query, the hidden state will turn from pure support FG features to the mixture of support and query features, i.e., the proportion of support features will gradually reduce because the size of hidden state is fixed. As a result, the support information cannot be sufficiently utilized by many query pixels, especially for those pixels at end; (2) *Intra-class gap*: Although query and support FG objects belong to the same class, they can still be visually different [2, 54]. Therefore, the query FG features may prefer not to fuse the support information from the hidden state, but focus more on themselves, thereby leading to ineffective FSS since the success of FSS relies on effective utilization of the support information.

To address the aforementioned issues, we follow two intuitive design principles and present a **hybrid Mamba block (HMB)** that includes a couple of Mambas for more effective support-query features fusion: (1) **Support recapped Mamba**: Due to the fact that *continuously scanning* query features

will gradually reduce the support FG features in the hidden state, as shown in Figure 1(b), we adopt a simple yet effective strategy to **periodically recap (i.e., re-scan)** support FG features when scanning query features. Specifically, we split query features into small patches, and downsample the support FG features to the same size as a patch. Then, we rearrange a sequence in the form of alternatively appeared support features and query patches. In this way, the support FG features can be regularly recalled, and each query patch can witness sufficient support information during the scan; (2) **Query intercepted Mamba**: To tackle the *intra-class gap* issue, we simply **intercept the mutual interactions among query pixels** when propagating the hidden state (with pure support FG) to each query FG pixel, so each query FG pixel would have no choice (i.e., inaccessible to other query FG) but to inevitably fuse support FG. As illustrated in Figure 1(c), the query pixels are scanned in parallel (for query-query interception), instead of the sequential scan in standard Mamba.

To the best of our knowledge, we are the first to introduce the efficient Mamba to FSS. Particularly, we indicate two issues suffered by the original Mamba when being applied to the cross attention case, namely, the *support forgetting* issue and the *intra-class gap* issue. To address them, we design a hybrid Mamba block (HMB), which can effectively incorporate query FG features with the support FG features, thereby leading to better FSS performance. Extensive experiments have been conducted on two public benchmark datasets PASCAL-$5^i$ and COCO-$20^i$, demonstrating the superiority of our design. Notably, our model can surpass existing state-of-the-arts by up to 2.2% and 3.2% on PASCAL-$5^i$ and COCO-$20^i$, in terms of mean intersection over union (mIoU).

## 2 Related Work

**Few-shot segmentation.** The success of FSS relies heavily on the effective use of support samples, based on which existing methods can be divided into prototypical methods [2, 16–18, 24, 29, 34, 39, 40, 45, 46, 57] and attention-based methods [11–13, 15, 22, 30, 31, 35, 44, 48, 50, 52–54, 58, 60, 64]. Prototypical methods compress support FG features into prototype(s) [18, 46], which are used to segment the query image through either feature comparisons [2, 46] or feature concatenation [40]. Notably, SSP [2] first introduces the intra-class gap issue, i.e., query FG tends to be more similar to itself rather than support FG. They use the support prototype to mine discriminative query features first, which are then used to find other similar query features, where the query-query matching will not suffer from the issue. The prototypes are obtained at the cost of information loss, so attention-based methods build up pairwise relationships between query and support pixels instead. Despite of their performance, attentions have quadratic complexity to the feature sizes, impeding the use of more attention blocks, and making the inference speed slow. In this paper, we propose to incorporate Mamba [5] into FSS, which has linear complexity but can also capture long-range dependencies.

**Mamba.** Mamba [5] improves state space models (SSMs) [3, 6, 27, 38] by introducing a selection mechanism to make the parameters input-dependent, and has shown appealing performance on language and speech tasks, with linear complexity. Inspired by this, researchers have applied Mamba to various vision tasks, and we divide existing methods into self [4, 7, 20, 25, 51, 55, 68] and cross Mamba [10, 19, 32, 43, 56, 61], for capturing the intra- and inter-sequence correlations, respectively. Self Mamba methods mainly focus on defining different scanning rules, e.g., vision Mamba [68] and VMamba [25] introduce bidirectional and 4-directional scans to better capture long-range dependencies for 2D images, some methods [7, 51, 55] define rules for 3D data, etc. Most cross Mamba methods study the scenario where different modalities have already been pixel-wise aligned, e.g., Sigma [43] exchanges the parameters of two Mambas that correspond to the aligned RGB and infrared images, Pan-Mamba [10] and CFMW [19] swap the different-modality features for a pixel. Besides, ReMamber [56] channel-wise concatenate two modalities, and perform a channel-wise Mamba to propagate the information from one modality to another. Unfortunately, the query and support images in FSS cannot be pixel-wise aligned, as they contain different objects. The only possible way [32, 61] is to flatten the query and support features into sequences, and concatenate them to be a longer sequence. Hence, the support features (the first half) can be gradually compressed into the hidden state, and be fused into query features when scanning the query features.

## 3 Problem Definition

FSS aims at segmenting objects with arbitrary classes, with a few support pairs that contain the same-class objects. To this end, a training paradigm called episodic training [42] is introduced, where

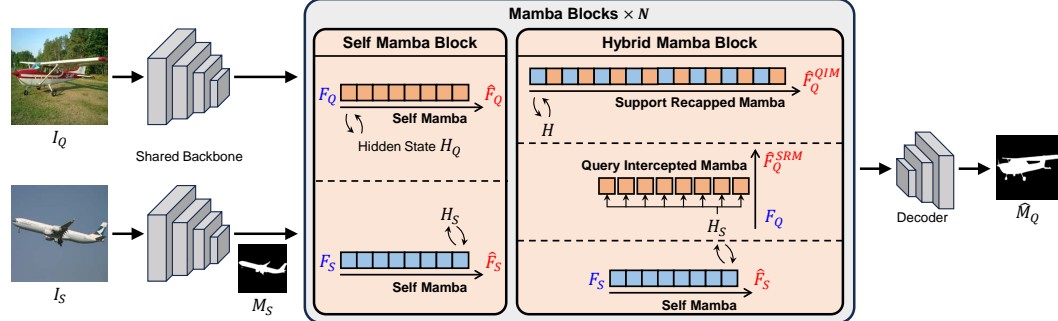

Figure 2: Overview of HMNet. Mamba blocks consist of alternatively appeared self Mamba blocks (SMB) and hybrid Mamba blocks (HMB). Self Mamba aims at capturing the intra-sequence correlations, while hybrid Mamba attempts to capture the support-query intra-sequence dependencies. Hybrid Mamba further includes a support recapped Mamba (SRM) and a query intercepted Mamba (QIM) to address the *support forgetting* and *intra-class gap* issues.

the dataset is split into a train set $\mathcal{D}_{\text{train}}$ and a test set $\mathcal{D}_{\text{test}}$, encompassing a series of episodes. Each episode comprises a query set $\mathcal{Q} = \{I_Q, M_Q\}$ and a support set $\mathcal{S} = \{I_S^k, M_S^k\}_{k=1}^K$ in the $K$-shot setting, where $I$ and $M$ represent the input images and the binary masks. Note that the classes involved in $\mathcal{D}_{\text{train}}$ form a base class set $\mathcal{C}_{\text{base}}$, and those in $\mathcal{D}_{\text{test}}$ form a novel class set $\mathcal{C}_{\text{novel}}$, FSS studies a scenario where $\mathcal{C}_{\text{base}} \cap \mathcal{C}_{\text{novel}} = \emptyset$. In brief, FSS would sample some episodes from $\mathcal{D}_{\text{train}}$ to learn the pattern of using support information to segment the query image, and directly applying the learned pattern to the episodes sampled from $\mathcal{D}_{\text{test}}$ for segmenting previously unseen classes. The methodology is described under 1-shot setting.

## 4 Methodology

### 4.1 Revisit Mamba

The essence of Mamba [5] is a structured state space model (SSM), which originates from the continuous system, i.e., linear time-invariant (LTI) system, that maps a 1D sequence from $x(t)$ to $y(t)$. Such mapping is achieved through an intermediate hidden state $h(t)$ and a linear ordinary differential equations (ODEs) [25] as follows:

$$h'(t) = \mathbf{A}h(t) + \mathbf{B}x(t)$$
$$y(t) = \mathbf{C}h(t) \tag{1}$$

where $A$ is a evolution parameter, $B$ and $C$ denote two projection parameters. Then, SSM employs a zero-order hold (ZOH) strategy to transform the contiguous system into a discrete one:

$$\bar{\mathbf{A}} = exp(\Delta\mathbf{A})$$
$$\bar{\mathbf{B}} = (\Delta\mathbf{A})^{-1}(exp(\Delta\mathbf{A}) - \mathbf{I})\Delta\mathbf{B} \tag{2}$$

where $\Delta$ denotes the timescale parameter, and Equation 1 can be rewritten as:

$$h_t = \bar{\mathbf{A}}h_{t-1} + \bar{\mathbf{B}}x_t$$
$$y_t = \mathbf{C}h_t \tag{3}$$

As described in Mamba [5], the discrete parameters $\bar{\mathbf{A}}$ and $\bar{\mathbf{B}}$ are constant dynamics, so they cannot effectively compress information into the hidden state and fuse correct information from the context, leading to the failure of capturing long-range dependencies. To this end, Mamba proposes to equip SSM with a selection mechanism (denoted as selective SSM) to make parameters $\mathbf{B}$, $\mathbf{C}$ and $\Delta$ input-dependent, which is validated to be capable of capturing complex correlations.

### 4.2 Hybrid Mamba Network (HMNet)

As shown in Figure 2, we present hybrid Mamba network (HMNet) to incorporate the efficient Mamba with FSS. Following existing FSS methods [16, 40, 48, 54], the query image $I_Q$ and the

support image $I_S$ are forwarded to a pretrained backbone, like VGG16 [37] or ResNet50 [9], to obtain the mid-level query and support features. Then, they are forwarded to some alternatively appeared self Mamba block (SMB) and hybrid Mamba block (HMB) for feature enhancement. Specifically, SMB aims at modeling the intra-sequence correlations for query and support features, while HMB (Section 4.2.1 aims to fuse sufficient support FG features into query FG features. Particularly, HMB further contains a support recapped Mamba (SRM) and a query intercepted Mamba (QIM), to mitigate the aforementioned *support forgetting* and *intra-class gap* issues (in Section 1). Finally, the enhanced query features are processed by a decoder [54] to obtain the predictions $\hat{M}_Q$.

Kindly remind that our contribution is to improve cross Mamba for capturing support-query inter-sequence dependencies, instead of improving self Mamba. Hence, we take VMamba [25] to build SMB, whose details are displayed in Figure 6 (in Appendix D.1). The 2D image features would be reshaped into 4 sequences, according to different scanning directions. Then, these sequences are scanned with separate Mambas (with distinct parameters) for feature enhancement. Finally, 4 sequences are reshaped back to 4 features, which are summed up to obtain the output features.

### 4.2.1 Hybrid Mamba Block (HMB)

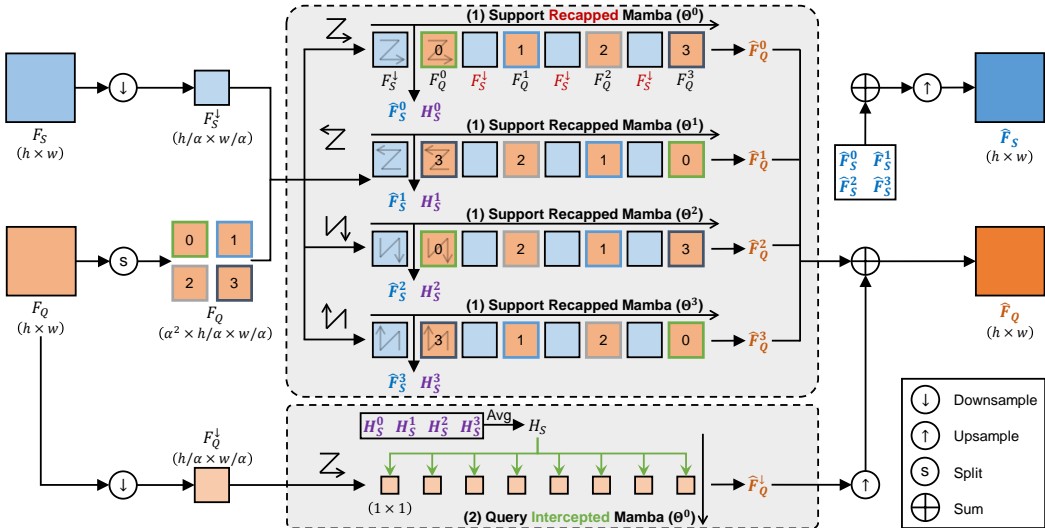

Figure 3: Illustration of HMB. (1) Based on different scanning directions [25], **SRM** arranges support and query features into 4 sequences in the form of alternatively appeared support and query patches, which are **sequentially** scanned with 4 sets of parameters $\Theta$. (2) After scanning support features for the first time in SRM, 4 hidden states are averaged into $H_S$. In **QIM**, $H_S$ is used to scan query features **in parallel**. Note that QIM's parameter is shared with the first SRM.

The details of HMB are shown in Figure 3, which is formally described as follows. For ease of illustration, we omit the batch size and hidden dimension when describing the shape of variables.

**Feature preparation.** Although query and support images share the same FG class, they usually have different BG classes, so it is a common practice to mask the support BG [16, 40], i.e., the support features are sparse. Some methods [31, 48] have validated that downsampling the support features to some extent will not decrease the performance, but can save much memory. Hence, we adopt the same strategy to obtain $F_S^\downarrow \in \mathbb{R}^{\frac{h}{\alpha} \times \frac{w}{\alpha}}$. Then, we make two copies of the query features: (1) For SRM, we keep the original granuarity for dense segmentation, but split the features into patches $F_Q \in \mathbb{R}^{\alpha^2 \times \frac{h}{\alpha} \times \frac{w}{\alpha}}$, where each patch has the same size as the downsampled support features; (2) For QIM, we downsample the query features as $F_Q^\downarrow \in \mathbb{R}^{\frac{h}{\alpha} \times \frac{w}{\alpha}}$, so its granuarity will be consistent with that of $F_S^\downarrow$ for better feature fusion.

$$F_S^\downarrow = \text{Down}(F_S, \alpha), F_Q^\downarrow = \text{Down}(F_Q, \alpha), F_Q = \text{Split}(F_Q, \alpha) \tag{4}$$

where $\alpha$ is the downsample/split ratio, and it is empirically set as 4 in this paper.

**SRM.** As shown in Figure 3(1), SRM is designed to **periodically recap** the support features during the scan on query. We first reshape the query patches $F_Q$ into 4 query sequences $L_Q^i \in \mathbb{R}^{\frac{hw}{\alpha^2} \times \alpha^2}, i \in [0, 3]$, with different scanning directions. Take $L_Q^0$ as an example:

$$L_Q^0 = (F_Q^0 || F_Q^1 || \cdots || F_Q^{\alpha^2 - 1}) \tag{5}$$

where $F_Q^j, j \in [0, \cdots, \alpha^2 - 1]$ denotes a reshaped query patch (the inner-patch pixel-wise scan direction is the same as patch-wise scan direction), $||$ means patch concatenation. Then, we repeat and **recap** $F_S^{\downarrow}$ by $\alpha^2$ times, and reshape them into 4 support sequences $L_S^i \in \mathbb{R}^{\frac{hw}{\alpha^2} \times \alpha^2}, i \in [0, 3]$:

$$L_S^0 = (F_S^{\downarrow} || F_S^{\downarrow} || \cdots || F_S^{\downarrow}) \tag{6}$$

Next, we concatenate the support and query sequences along the pixel dimension as:

$$L_{SRM}^i = (L_S^i || L_Q^i) \tag{7}$$

where $L_{SRM}^i \in \mathbb{R}^{\frac{2hw}{\alpha^2} \times \alpha^2}$ denote $\alpha^2$ pairs of support and query patches. Support is put ahead of query because we aim to propagate the former to the latter. Thereafter, $L_{SRM}^i$ are flattened to 1D sequences, and we use 4 sets of Mamba parameters $\Theta^i$ to scan the sequences with Equation 3:

$$\hat{F}_Q^i, (\hat{F}_S^i, H_S^i) = \text{SRM}(L_{SRM}^i, \Theta^i) \tag{8}$$

where $\hat{F}_Q^i \in \mathbb{R}^{h \times w}$ are enhanced query features, detached from the sequences and then reshaped. $\hat{F}_S^i \in \mathbb{R}^{\frac{h}{\alpha} \times \frac{w}{\alpha}}$ are the reshaped support features taken from the head of the sequences, we do not consider support features in other positions, as they have already been mingled with query features. $H_S^i$ is the hidden state obtained from the $i$th sequence after scanning the first support features.

**QIM.** To encourage query features to fuse more support features, we further design a QIM (in Figure 3(2)) to intercept the mutual interactions among query pixels, and propagate the hidden state to each query pixel **in parallel**. Hence, there is no need to reshape query pixels into multiple sequences based on different directions. We directly flatten $F_Q^{\downarrow}$ into a 1D sequence, and take the averaged hidden states (with pure support features) obtained from SRM as the initial hidden state for QIM:

$$H_S = \text{Avg}(H_S^i), i \in [0, 3] \tag{9}$$

Then, Equation 3 can be directly calculated with matrix multiplication and rewritten as:

$$\hat{F}_Q^{\downarrow} = \text{QIM}(F_Q^{\downarrow}, H_S, \Theta^0) = \mathbf{C}\bar{\mathbf{A}}H_S + \mathbf{C}\bar{\mathbf{B}}F_Q^{\downarrow} \tag{10}$$

where $\mathbf{C}, \bar{\mathbf{A}}$ and $\bar{\mathbf{B}}$ form a set of Mamba parameters. Equation 10 can be interpreted as a special case of Equation 3, where the length of each sequence is 1. To facilitate better parameters learning, we **share the parameters $\Theta^0$ of SRM with QIM**, which are learned with long sequences.

**Feature ensemble.** At last, we use sum fusion to obtain the output features of hybrid Mamba:

$$\hat{F}_Q = \text{Sum}(\hat{F}_Q^i) + \text{Up}(\hat{F}_Q^{\downarrow}, \alpha)$$
$$\hat{F}_S = \text{Up}(\text{Sum}(\hat{F}_S^i), \alpha) \tag{11}$$

where $\text{Up}(\cdot)$ means upsampling features to the original size.

### 4.2.2 Computational Complexity

We follow Vim [68] to analyze the computational complexity. Given an input sequence $L \in \mathbb{R}^{M \times D}$, the complexity of attention [41], Mamba [5] (1 direction) and VMamba [25] (4 directions) are:

$$\Omega(\text{Attention}) = 4MD^2 + 2M^2D$$
$$\Omega(\text{Mamba}) = 8MDN \tag{12}$$
$$\Omega(\text{VMamba}) = 4 \times \Omega(\text{Mamba}) = 32MDN$$

where $M = h \times w$ indicates the length of the sequence, $D$ is the hidden dimension, $N$ is a small constant (e.g., 16), denoting the size of hidden state. In hybrid Mamba, SRM assembles query and support sequences into 4 $L_{SRM}^i \in \mathbb{R}^{2M \times D}$ (4 directions, longer sequence), while QIM deals with flattened $F_Q^{\downarrow} \in \mathbb{R}^{\frac{M}{\alpha^2} \times D}$ (1 direction, shorter sequence), so the total complexity is:

$$\Omega(\text{Hybrid Mamba}) = 2 \times \Omega(\text{VMamba}) + \Omega(\text{Mamba})/\alpha^2 = (64 + 8/\alpha^2)MDN \tag{13}$$

where $\alpha$ is the downsample ratio (e.g., 4), and the complexity is still linear to the sequence length.

# 5 Experiments

## 5.1 Experiment Setup

**Datasets.** The methods are evaluated on two benchmark datasets, including PASCAL-$5^i$ [34] and COCO-$20^i$ [28]. PASCAL-$5^i$ is built on top of PASCAL VOC 2012 [1], with additional annotations obtained from SBD [8], while COCO-$20^i$ is created from MSCOCO dataset [21]. PASCAL-$5^i$ comprises 20 distinct classes, and COCO-$20^i$ is a larger benchmark that includes 80 classes. Following existing works [40, 46, 59], both of them are evenly split into four folds based on the classes, and each fold would consist of 5 and 20 classes for PASCAL-$5^i$ and COCO-$20^i$, respectively. Then, cross validations are carried out, with each fold being taken as the test set once, while the union of other folds is adopted for training. In the test phase, 1,000 (for PASCAL-$5^i$) and 4,000 (for COCO-$20^i$) episodes are randomly sampled to comprehensively evaluate the performance of various models.

**Evaluation metrics.** Following existing baselines [16, 31, 40, 48, 54], mean intersection over union (mIoU) and foreground-background IoU (FB-IoU) are deployed as the evaluation metrics.

## 5.2 Comparisons with State-of-the-Arts

Table 1: Performance comparisons with state-of-the-arts on PASCAL-$5^i$. "$5^i$" denotes the mIoU score of the $i$-th fold, "Mean" is the averaged mIoU score of 4 folds, "FB-IoU" is averaged from 4 folds. **Bold** values show the best performance.

| Backbone | Method | 1-shot | | | | | | 5-shot | | | | | |
|---|---|---|---|---|---|---|---|---|---|---|---|---|---|
| | | $5^0$ | $5^1$ | $5^2$ | $5^3$ | mIoU | FB-IoU | $5^0$ | $5^1$ | $5^2$ | $5^3$ | mIoU | FB-IoU |
| VGG16 | PFENet (TPAMI'20) [40] | 56.9 | 68.2 | 54.4 | 52.5 | 58.0 | 72.0 | 59.0 | 69.1 | 54.8 | 52.9 | 59.0 | 72.3 |
| | DACM (ECCV'22) [52] | 61.8 | 67.8 | 61.4 | 56.3 | 61.8 | 75.5 | 66.1 | 70.6 | 65.8 | 60.2 | 65.7 | 77.8 |
| | FECANet (TMM'23) [22] | 66.5 | 68.9 | 63.6 | 58.3 | 64.3 | 76.2 | 68.6 | 70.8 | 66.7 | 60.7 | 66.7 | 77.6 |
| | SCCAN (ICCV'23) [54] | 63.3 | 70.8 | 66.6 | 58.2 | 64.7 | 77.2 | 67.2 | 72.3 | 70.5 | 63.8 | 68.4 | 79.1 |
| | BAM (CVPR'22) [16] | 63.2 | 70.8 | 66.1 | 57.5 | 64.4 | 77.3 | 67.4 | 73.1 | 70.6 | 64.0 | 68.8 | 81.1 |
| | SVF (NIPS'22) [39] | 64.1 | 71.1 | 66.8 | 57.5 | 64.9 | - | 67.8 | 74.1 | 71.0 | 63.6 | 69.1 | - |
| | HDMNet (CVPR'23) [31] | 64.8 | 71.4 | 67.7 | 56.4 | 65.1 | - | 68.1 | 73.1 | 71.8 | 64.0 | 69.3 | - |
| | HMNet (ours) | **66.7** | **74.5** | **68.9** | **59.0** | **67.3** | **79.2** | **70.5** | **76.0** | **72.2** | **65.7** | **71.1** | **82.6** |
| ResNet50 | PFENet (TPAMI'20) [40] | 61.7 | 69.5 | 55.4 | 56.3 | 60.8 | 73.3 | 63.1 | 70.7 | 55.8 | 57.9 | 61.9 | 73.9 |
| | CyCTR (NIPS'21) [60] | 67.8 | 72.8 | 58.0 | 58.0 | 64.2 | - | 71.1 | 73.2 | 60.5 | 57.5 | 65.6 | - |
| | VAT (ECCV'22) [11] | 67.6 | 72.0 | 62.3 | 60.1 | 65.5 | 77.8 | 72.4 | 73.6 | 68.6 | 65.7 | 70.1 | 80.9 |
| | DACM (ECCV'22) [52] | 66.5 | 72.6 | 62.2 | 61.3 | 65.7 | 77.8 | 72.4 | 73.7 | 69.1 | 68.4 | 70.9 | 81.3 |
| | ABCNet (CVPR'23) [49] | 68.8 | 73.4 | 62.3 | 59.5 | 66.0 | 76.0 | 71.7 | 74.2 | 65.4 | 67.0 | 69.6 | 80.0 |
| | SCCAN (ICCV'23) [54] | 68.3 | 72.5 | 66.8 | 59.8 | 66.8 | 77.7 | 72.3 | 74.1 | 69.1 | 65.6 | 70.3 | 81.8 |
| | FECANet (TMM'23) [22] | 69.2 | 72.3 | 62.4 | **65.7** | 67.4 | 78.7 | 72.9 | 74.0 | 65.2 | 67.8 | 70.0 | 80.7 |
| | BAM (CVPR'22) [16] | 69.0 | 73.6 | 67.6 | 61.1 | 67.8 | 79.7 | 70.6 | 75.1 | 70.8 | 67.2 | 70.9 | 82.2 |
| | SVF (NIPS'22) [39] | 69.4 | 74.5 | 68.8 | 63.1 | 69.0 | 80.1 | 72.1 | 76.2 | 72.0 | 68.9 | 72.3 | 83.2 |
| | HDMNet (CVPR'23) [31] | 71.0 | 75.4 | 68.9 | 62.1 | 69.4 | - | 71.3 | 76.2 | 71.3 | 68.5 | 71.8 | - |
| | AMNet (NIPS'23) [48] | 71.1 | **75.9** | 69.7 | 63.7 | 70.1 | - | 73.2 | **77.8** | 73.2 | 68.7 | 73.2 | - |
| | HMNet (ours) | **72.2** | 75.4 | **70.0** | 63.9 | **70.4** | **81.6** | **74.2** | 77.3 | **74.1** | **70.9** | **74.1** | **84.4** |

**Quantitative results.** The quantitative comparisons between the proposed HMNet and the state-of-the-arts are presented in Table 1 and Table 2 for PASCAL-$5^i$ and COCO-$20^i$, respectively, and we could observe that HMNet can outperform existing methods by considerable margins in almost all cases. For instance, with VGG16 as the backbone, HMNet can surpass HDMNet by 1.8% in terms of mean mIoU under both 1-shot and 5-shot settings. Besides, the 1-shot and 5-shot FB-IoU scores can be as high as 81.6% and 84.4%, when ResNet50 is taken as the backbone. Notably, we observe that the performance gain appears to be larger on COCO-$20^i$, e.g., 3.2% (VGG16, 1-shot) and 1.6% (ResNet50, 5-shot). We attribute it to the fact that COCO-$20^i$ is a more complicated dataset, compared to PASCAL-$5^i$, e.g., the image samples not only have more complex background, but the *intra-class gap* issue tends to be much more severer, while the designed QIM can help to address this issue well, i.e., encourage query features to fuse more support features.

**Qualitative results.** We pick some episodes from PASCAL-$5^i$ and COCO-$20^i$, then visually compare one of the best baselines, HDMNet [31], with our HMNet in Figure 4. We can observe that our model is better at distinguishing the query FG and BG objects than HDMNet, e.g., HDMNet would sometimes (1) wrongly classify query BG objects as FG (column 1, 2, 3, 5, 8); and (2) fail to ensure the integrity of the predicted FG objects (column 4, 5, 7). We refer to the BG mismatch and FG-BG entanglement issues [54], mainly raised by the fact that query BG cannot find matched features from support FG, for the possible reasons to the failure of HDMNet. Instead, (1) in our hybrid Mamba

Table 2: Performance comparisons with state-of-the-arts on COCO-$20^i$. "$20^i$" denotes the mIoU score of the $i$-th fold, "Mean" is the averaged mIoU score of 4 folds, "FB-IoU" is averaged from 4 folds. **Bold** values show the best performance.

| Backbone | Method | 1-shot $20^0$ | $20^1$ | $20^2$ | $20^3$ | mIoU | FB-IoU | 5-shot $20^0$ | $20^1$ | $20^2$ | $20^3$ | mIoU | FB-IoU |
|---|---|---|---|---|---|---|---|---|---|---|---|---|---|
| VGG16 | PFENet (TPAMI'20) [40] | 35.4 | 38.1 | 36.8 | 34.7 | 36.3 | 63.3 | 38.2 | 42.5 | 41.8 | 38.9 | 40.4 | 65.0 |
| | FECANet (TMM'23) [22] | 34.1 | 37.5 | 35.8 | 34.1 | 35.4 | 65.5 | 39.7 | 43.6 | 42.9 | 39.7 | 41.5 | 67.7 |
| | SCCAN (ICCV'23) [54] | 38.3 | 46.5 | 43.0 | 41.5 | 42.3 | 66.9 | 43.4 | 52.5 | 54.5 | 47.3 | 49.4 | 71.8 |
| | BAM (CVPR'22) [16] | 39.0 | 47.0 | 46.4 | 41.6 | 43.5 | - | 47.0 | 52.6 | 48.6 | 49.1 | 49.3 | - |
| | SVF (NIPS'22) [39] | 40.2 | 46.6 | 46.2 | 42.0 | 43.8 | - | 45.1 | 53.6 | 48.4 | 49.3 | 49.1 | - |
| | HDMNet (CVPR'23) [31] | 40.7 | 50.6 | 48.2 | 44.0 | 45.9 | - | 47.0 | 56.5 | 54.1 | 51.9 | 52.4 | - |
| | HMNet (ours) | **44.2** | **51.8** | **51.9** | **48.4** | **49.1** | **72.6** | **48.8** | **58.0** | **57.9** | **53.2** | **54.5** | **75.5** |
| ResNet50 | PFENet (TPAMI'20) [40] | 36.5 | 38.6 | 35.0 | 33.8 | 35.8 | - | 36.5 | 43.3 | 38.0 | 38.4 | 39.0 | - |
| | CyCTR (NIPS'21) [60] | 38.9 | 43.0 | 39.6 | 39.8 | 40.3 | - | 41.1 | 48.9 | 45.2 | 47.0 | 45.6 | - |
| | VAT (ECCV'22) [11] | 39.0 | 43.8 | 42.6 | 39.7 | 41.3 | 68.8 | 44.1 | 51.1 | 50.2 | 46.1 | 47.9 | 72.4 |
| | DACM (ECCV'22) [52] | 37.5 | 44.3 | 40.6 | 40.1 | 40.6 | 68.9 | 44.6 | 52.0 | 49.2 | 46.4 | 48.1 | 71.6 |
| | ABCNet (CVPR'23) [49] | 42.3 | 46.2 | 46.0 | 42.0 | 44.1 | 69.9 | 45.5 | 51.7 | 52.6 | 46.4 | 49.1 | 72.7 |
| | FECANet (TMM'23) [22] | 38.5 | 44.6 | 42.6 | 40.7 | 41.6 | 69.6 | 44.6 | 51.5 | 48.4 | 45.8 | 47.6 | 71.1 |
| | SCCAN (ICCV'23) [54] | 40.4 | 49.7 | 49.6 | 45.6 | 46.3 | 69.9 | 47.2 | 57.2 | 59.2 | 52.1 | 53.9 | 74.2 |
| | BAM (CVPR'22) [16] | 43.4 | 50.6 | 47.5 | 43.4 | 46.2 | - | 49.3 | 54.2 | 51.6 | 49.6 | 51.2 | - |
| | SVF (NIPS'22) [39] | **46.9** | 53.8 | 48.4 | 44.8 | 48.5 | - | 52.3 | 57.8 | 52.0 | 53.4 | 53.9 | - |
| | HDMNet (CVPR'23) [31] | 43.8 | 55.3 | 51.6 | 49.4 | 50.0 | 72.2 | 50.6 | 61.6 | 55.7 | 56.0 | 56.0 | 77.7 |
| | AMNet (NIPS'23) [48] | 44.9 | 55.8 | 52.7 | 50.6 | 51.0 | 72.9 | 52.0 | 61.9 | 57.4 | **57.9** | 57.3 | **78.8** |
| | HMNet (ours) | 45.5 | **58.7** | 52.9 | 51.4 | **52.1** | **74.5** | **53.4** | **64.6** | **60.8** | 56.8 | **58.9** | 77.6 |

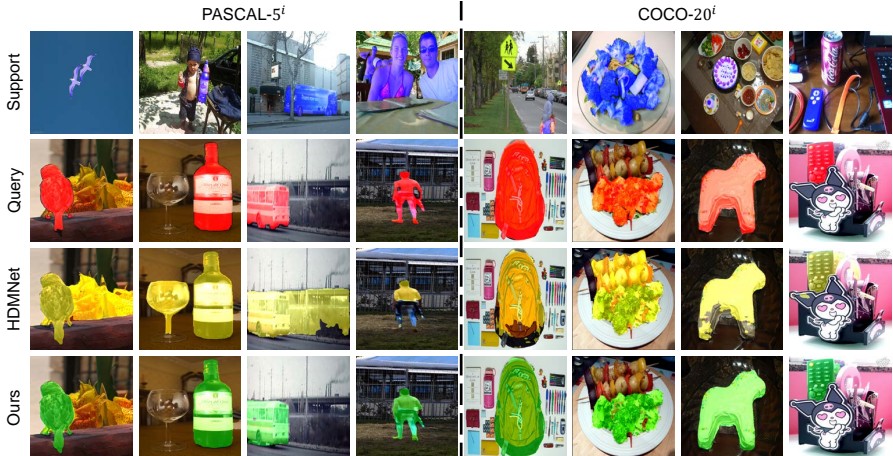

Figure 4: Qualitative comparisons with HDMNet [31] on PASCAL-$5^i$ and COCO-$20^i$.

block, both query FG and BG can be fused with matched features, thus, these issues [54] would not arise; and (2) the proposed SRM and QIM can help to effectively utilize the support FG information, leading to better FSS performance. We include more qualitative results in Appendix 8, 9 and 10.

## 5.3 Ablation Study

Table 3: Component-wise ablation study.

| SMB | SRM | QIM | BAM | $5^0$ | $5^1$ | $5^2$ | $5^3$ | mIoU |
|---|---|---|---|---|---|---|---|---|
| | | | | 65.6 | 71.5 | 64.6 | 59.2 | 65.2 |
| ✓ | | | | 68.8 | 72.6 | 67.6 | 60.3 | 67.4 |
| ✓ | Basic | | | 68.6 | 72.6 | 67.0 | 62.0 | 67.5 |
| ✓ | Recap | | | 70.1 | 73.5 | 67.1 | 61.6 | 68.1 |
| ✓ | | Basic | | 69.1 | 73.4 | 67.9 | 60.9 | 67.8 |
| ✓ | | Share | | 70.0 | 73.4 | 67.4 | 61.7 | 68.1 |
| ✓ | Recap | Share | | 70.1 | 73.6 | 67.7 | 62.9 | 68.6 |
| ✓ | Recap | Share | ✓ | 72.2 | 75.4 | 70.0 | 63.9 | 70.4 |

Table 4: Parameter study on the number of Mamba blocks, under 1-shot setting. FLOPs are measured with query and support images whose shapes are 473×473×3.

| #Blocks | mIoU | #Params | #FLOPs | FPS |
|---|---|---|---|---|
| 4 | 67.8 | 32.8M | 449.4G | 11.9 |
| 6 | 68.2 | 35.0M | 466.8G | 10.3 |
| 8 | 68.6 | 37.1M | 484.3G | 9.6 |
| 10 | 67.5 | 39.2M | 501.8G | 8.5 |

**Component-wise ablation.** We present the detailed component-wise ablation study in Table 3 to validate the effectiveness of different modules. To have better comparisons with attention-based

methods, we start from a basic model tailored from SCCAN [54], with all attention-related modules removed, and the mIoU score is initialized as 65.2%. When we merely add self Mamba blocks (SMB) into the basic model, the mIoU score has already been boosted to 67.4%, showing the superiority of Mamba [5]. Then, we further incorporate hybrid Mamba blocks (HMB) to capture the intra-sequence dependencies. An HMB consists of a pair of support recapped Mamba (SRM) and query intercepted Mamba (QIM). For SRM, "Basic" means Equation 6 is not performed, and we sequentially scan the support and query features to achieve cross Mamba. Nevertheless, the performance gain is only 0.1%, for which we explain as the *support forgetting* issue. When we periodically recap the support features during the scan on query, the mIoU score can increase to 68.1%. As for QIM, the "Basic" version corresponds to the case where QIM's Mamba parameter $\Theta$ is not shared with the first support features in SRM. From the table, we could observe that parameter sharing can improve the performance by 0.3%. After that, we employ both SRM and QIM, and the mIoU score can be as high as 68.6%. Following [31, 48], we finally ensemble our model with BAM's finetuned backbones and base class annotations to boost the score to 70.4%.

Also note that under the same condition, our model can outperform attention-based SCCAN by 1.8% (68.6% v.s. 66.8%), while the computational cost is lower, demonstrating the proposed Mamba-based modules can be both more effective and efficient than previous attention-based modules.

**Parameter study on block number.** Kindly remind that we alternatively employ SMB and HMB for capturing intra- and inter-sequence dependencies. To show the impacts of different number of Mamba blocks, we employ 4, 6, 8 and 10 blocks for experiments, and show the results in Table 4.Note that the parameter number and floating point operations (FLOPs) cover all modules, including the pretrained backbone. We can observe that (1) the best performance can be achieved when the number of blocks is 8, and (2) each pair of SMB and HMB only has about 2M parameters, and the corresponding FLOPs are approximately 18G, proving that the proposed Mamba blocks are cost-effective.

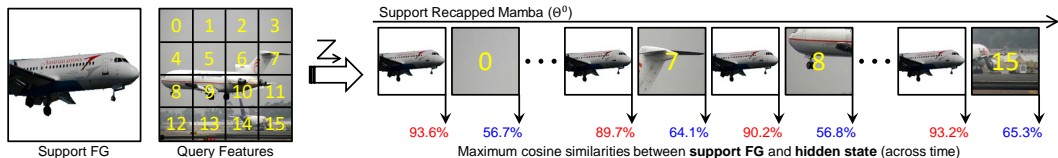

Figure 5: Changes of the hidden state in SRM across time, take the first SRM as an example.

**Visualization of SRM.** Existing methods sequentially scan the support and query sequences, to incorporate the former to the latter. However, they will suffer from the *support forgetting* issue, where the support features in hidden state will gradually be replaced as query features. To tackle this, we design SRM, and provide an visual example in Figure 5. Recall that query features will be split into patches, and support features will be downsampled to the same size as a query patch. Then, a sequence of alternatively appeared support FG features and query patches will be scanned by SRM for feature enhancement. To prove the existence of the issue, as well as the effectiveness of SRM, we obtain the hidden states after scanning each support or query patch, and apply the cosine similarity operator to measure the maximum similarity between support pixels and the hidden states. We could observe: (1) The *support forgetting* issue really exists, as the blue scores are much smaller than the red scores; (2) Our mechanism of periodically recapping the support features, though simple, is effective, because the small values can be brought back to larger ones consistently.

## 6  Conclusion

In this paper, we first incorporate Mamba into FSS to capture inter-sequence dependencies. A simple way is to scan the support sequence first, then use its last hidden state to start scanning query sequence. However, this method suffers from *support forgetting* and *intra-class gap* issues. To alleviate them, we design a hybrid Mamba block (HMB) that further contains a support recapped Mamba (SRM) and a query intercepted Mamba (QIM). SRM will periodically recap the support features when scanning on query features, leading to better utilization of support information. Besides, QIM will intercept the mutual interaction of query pixels, so they can fuse more information from support features, instead of themselves. Extensive experiments have been conducted to validate the effectiveness of HMB.

**Acknowledgement.** This study is supported under the RIE2020 Industry Alignment Fund - Industry Collaboration Projects (IAF-ICP) Funding Initiative, as well as cash and in-kind contribution from the industry partner(s).

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

# A    Code

The source code is provided in the supplementary material. The instructions for obtaining the datasets, pretrained backbones and our trained models are detailed in the README file.

# B    Implementation Details

VGG16 [37] and ResNet50 [9] are deployed as the pretrained backbones. Then, we target two FSS settings in this paper, namely, with and without BAM's ensemble [16]. Specifically, BAM proposes to (1) finetune the previous backbones with $\mathcal{C}_{\text{base}}$, and (2) also use base classes' predictions during testing. We use AdamW to optimize Mamba-related parameters [5], and SGD to optimize the remaining parameters (e.g., decoder), with their learning rates initialized as 6e-5 and 5e-3. Following AMNet [48] and AENet [53], the model is trained for 300 epochs on PASCAL-$5^i$, and 75 epochs on COCO-$20^i$, with batch size set as 8 and 16, respectively. We enable DDP for model training, e.g., use 4 and 8 NVIDIA V100 GPUs for two datasets. Automatic mixed precision is enabled to make the training on COCO-$20^i$ faster. During training, all images are randomly cropped to $473 \times 473$ and $633 \times 633$ for PASCAL-$5^i$ and COCO-$20^i$, and we employ the same set of data augmentation techniques as [40]. We employ 8 Mamba blocks (i.e., 4 self and hybrid Mamba pairs), and set the hidden dimension as 256. Other Mamba-related hyperparameters are set as the same as [25]. For $K$-shot setting, when $K > 1$, we follow [40, 54] to average the support features.

# C    Additional Experiments

In this section, we provide some additional experiments to show the superiority of the proposed HMNet, including the use of more cost-effective weak support labels (Section C.1), the performance with different random seeds on COCO-$20^i$ (Section C.2), the performance with different number of testing episodes on COCO-$20^i$ (Section C.3), as well as the efficiency of HMNet (Section C.3).

## C.1    Weak Support Labels

Table 5: Study on weak support annotations. "Mask" represents pixel-wise support masks, and "Bounding box" means only drawing bounding boxes to wrap the support FG objects, whose annotation cost is much smaller.

| Method | Label | $5^0$ | $5^1$ | $5^2$ | $5^3$ | mIoU | FB-IoU |
|--------|-------|-------|-------|-------|-------|------|--------|
| | | | 1-shot | | | | |
| HMNet | Mask | 72.2 | 75.4 | 70.0 | 63.9 | 70.4 | 81.6 |
| | Bounding box | $71.6_{0.6\downarrow}$ | $74.5_{0.9\downarrow}$ | $68.8_{1.2\downarrow}$ | $62.1_{1.8\downarrow}$ | $69.2_{1.2\downarrow}$ | $80.7_{0.9\downarrow}$ |

Following existing methods [23, 46, 54, 59], we also experiment with the scenario where only weak support labels, e.g., bounding boxes, are provided. Generally, the time cost for annotating with weak labels is usually much smaller than that of the precise pixel-wise masks. Therefore, the cost for manual attention can be further reduced. The results are shown in Table 5, and it could be observed that the performance drop raised by weak support labels is minor, showing the effectiveness of the proposed HMNet.

## C.2    Error Bars Evaluation

To show the robustness towards different random seeds, we compare the 1-shot performance of HDMNet [31] and our model on COCO-$20^i$ dataset. Specifically, we use random seeds 0, 1, 2, 3 and 321 (default seed) to randomly sample 4,000 testing episodes from COCO-$20^i$, where the query samples, as well as their support samples, would be different. Note that COCO-$20^i$ is a challenging dataset, as (1) there exist many small objects, of which the mIoU scores tend to be small, and (2) the image samples are quite complicated, e.g., multiple objects, complex background, etc. Hence, the scores with different random seeds are likely to differ much. The testing results are displayed in Table 6, and we can could observe: (1) Our HMNet can consistently outperform HDMNet by large margins in all situations. Notably, our mIoU score averaged from 5 random seeds is 52.3%,

Table 6: Error bars evaluation with ResNet50 on COCO-20$^i$, under 1-shot setting. The random seeds are taken from {0, 1, 2, 3, 321} to generate 4,000 testing episodes. **Bold** values denote the best cases.

| Method | Seed | 1-shot | | | | | |
| | | 20$^0$ | 20$^1$ | 20$^2$ | 20$^3$ | mIoU | FB-IoU |
|---|---|---|---|---|---|---|---|
| HDMNet [31] | 0 | 45.5 | 55.3 | 49.6 | 46.7 | 49.3 | 71.9 |
| | 1 | 45.3 | 54.9 | 50.8 | 48.3 | 49.8 | 71.8 |
| | 2 | 44.9 | 54.2 | 50.0 | 48.7 | 49.5 | 72.2 |
| | 3 | 44.1 | 54.9 | 51.9 | 48.6 | 49.9 | 72.1 |
| | 321 | 44.8 | 54.9 | 50.0 | 48.7 | 49.6 | 72.0 |
| | Mean | 44.9 | 54.9 | 50.5 | 48.2 | 49.6 | 72.0 |
| | Std | **0.6** | 0.4 | 0.9 | 0.9 | 0.3 | 0.2 |
| HMNet (Ours) | 0 | 45.6 | 58.8 | 52.9 | 49.9 | 51.8 | 74.4 |
| | 1 | 47.4 | 58.2 | 53.1 | 51.3 | 52.5 | 74.6 |
| | 2 | 47.0 | 58.4 | 53.7 | 50.3 | 52.3 | 74.6 |
| | 3 | 47.2 | 58.6 | 54.5 | 50.0 | 52.6 | 74.8 |
| | 321 | 45.5 | 58.7 | 52.9 | 51.4 | 52.1 | 74.5 |
| | Mean | **46.5** | **58.5** | **53.4** | **50.6** | **52.3** | **74.6** |
| | Std | 0.9 | **0.2** | **0.7** | **0.7** | **0.3** | **0.1** |

2.7% better than HDMNet. In addition, the FB-IoU scores of HDMNet and our model are 72.0% and 74.6%, respectively; (2) In almost all cases, our model appears to be more robust than HMNet, i.e., the standard deviation (std) values of HMNet are smaller than those of HDMNet. To summarize, our model not only has good performance, but could also ensure the robustness towards randomness, showing the superiority of our module designs.

## C.3 Different Number of Testing Episodes

Table 7: Performance on COCO-20$^i$ with different number of testing episodes. The experiments are conducted with ResNet50, under 1-shot setting. The number of episodes are chosen from {4,000, 10,000, 20,000, 40,000}. **Bold** values denote the best cases.

| Method | #Episode | 1-shot | | | | | | |
| | | 20$^0$ | 20$^1$ | 20$^2$ | 20$^3$ | mIoU | FB-IoU | Avg. time (s) |
|---|---|---|---|---|---|---|---|---|
| HDMNet [31] | 4,000 | 44.8 | 54.9 | 50.0 | 48.7 | 49.6 | 72.0 | 681.2 |
| | 10,000 | 43.9 | 55.1 | 49.0 | 48.5 | 49.1 | 71.9 | 1806.2 |
| | 20,000 | 44.0 | 54.7 | 49.7 | 48.6 | 49.3 | 71.9 | 3555.4 |
| | 40,000 | 43.5 | 54.9 | 48.8 | 48.6 | 48.9 | 71.8 | 6728.0 |
| | Mean | 44.1 | 54.9 | 49.4 | 48.6 | 49.2 | 71.9 | 3192.7 |
| HMNet (Ours) | 4,000 | 45.5 | 58.7 | 52.9 | 51.4 | 52.1 | 74.5 | 578.0 |
| | 10,000 | 44.0 | 58.6 | 52.8 | 51.4 | 51.7 | 74.4 | 1438.1 |
| | 20,000 | 44.5 | 58.3 | 53.1 | 51.5 | 51.8 | 74.4 | 2920.4 |
| | 40,000 | 44.1 | 58.4 | 52.2 | 51.0 | 51.4 | 74.3 | 5728.5 |
| | Mean | **44.5** | **58.5** | **52.7** | **51.3** | **51.8** | **74.4** | **2666.3** |

In this section, we further test some models with different number of testing episodes on COCO-20$^i$. As displayed in Table 7, we not only provide with the testing performance, but also show the average time cost for testing one fold, It can be observed from the table: (1) Our model consistently performs better than HDMNet; (2) There is no prominent performance drop when testing on more episodes; (3) Kindly remind that HDMNet is an attention based method, while our HMNet is built based on Mamba. HDMNet employs 6 attention blocks, while our HMNet utilizes 8 Mamba blocks. Besides, the hidden dimensions of HDMNet and our model are 64 and 256, respectively. Moreover, HDMNet would downsample features with a ratio of 4 to calculate cross attention (i.e., the pixel number is reduced by 16 times), while we mainly conduct Mamba with the concatenation of query and support features (i.e., the pixel number is doubled). In brief, our model uses more blocks, higher hidden dimensions, and larger features sizes, but our HMNet still costs much less than HDMNet for testing, e.g., the average time costs for testing are 2,666.3 and 3,192.7 seconds for our model and HDMNet, showing that our HMNet is effective and efficient, and Mamba can be a good alternative to attention.

## C.4 Different Ways of Feature Fusion

Recall that we propose HMNet to employ memory-efficient Mamba to fuse support FG features into query features and activate query FG features accordingly. Alternatively, previous methods also use feature concatenation [16] and cross attention [54, 60] to achieve this goal. Specifically, (1) feature concatenation methods would first extract support FG features and compress it into prototype(s). Then, the prototype(s) would be expanded and concatenated with query features for further processing; (2) cross attention methods would measure the similarity between each pair of query and support pixels, based on which the support FG features will be dynamically aggregated into query features.

Table 8: Performance on PASCAL-$5^i$ with different feature fusion methods. The experiments are conducted with ResNet50, under 1-shot setting. **Bold** values denote the best cases.

| Method | 1-shot | | | | |
|---|---|---|---|---|---|
| | $5^0$ | $5^1$ | $5^2$ | $5^3$ | mIoU |
| BAM [16] | 65.6 | 71.5 | 64.6 | 59.2 | 65.2 |
| CyCTR [60] | 67.0 | 71.5 | 65.9 | 57.1 | 65.4 |
| SCCAN [54] | 68.3 | 72.5 | 66.8 | 59.8 | 66.8 |
| HMNet (Ours) | **70.1** | **73.6** | **67.7** | **62.9** | **68.8** |

Note that the values of BAM [16], CyCTR [60] and SCCAN [54] are reproduced under the same condition. Specifically, (1) we remove the unfair ensembles of BAM [16], and re-train the feature concatenation model; (2) CyCTR [60] designs the cycle-consistent attention, and we fairly reproduce its results; (3) the comparisons with SCCAN [54] is the fairest, which designs a self-calibrated cross attention. Particularly, we both use 8 attention/Mamba blocks, hidden dimensions, etc.

From Table 8, we can observe (1) our model behaves much better than others; (2) in the fairest case, our Mamba method is more effective than previous feature fusion methods, showing the superiority of our design.

## C.5 Efficiency Comparisons with Different Image Sizes

Recently, Mamba is well known for its ability to capture long-range dependencies as attention, while the memory cost is only linear to the number of pixels. Therefore, we further conduct experiments to show the effectiveness and efficiency of our Mamba-based HMNet.

Specifically, we interpolate COCO-$20^i$'s testing episodes to different sizes of 473×473, 633×633, 793×793, 953×953, 1113×1113, and the extracted feature sizes would be of 60×60, 80×80, 100×100, 120×120, 140×140. We compare our method with the cross attention-based HDMNet [31], and the time costs (seconds) for testing 4,000 episodes (ResNet50, 1-shot, with single 32GB V100 GPU) are shown in Table 9.

Table 9: Efficiency comparisons (seconds) on COCO-$20^i$ with different image sizes. The experiments are conducted with ResNet50, under 1-shot setting. **Bold** values denote the best cases. "OOM" means out-of-memory.

| Method | Image sizes | | | | |
|---|---|---|---|---|---|
| | 473×473 | 633×633 | 793×793 | 953×953 | 1113×1113 |
| HDMNet [31] | 415.4 | 676.5 | 1113.4 | 1713.7 | OOM |
| HMNet (Ours) | **402.4** | **576.9** | **768.9** | **1125.5** | **1440.6** |
| Difference | 13.0 | 99.6 | 344.5 | 588.2 | - |

Note that (1) HDMNet employs 6 attention blocks, while we use 8 Mamba blocks; (2) the hidden dimensions of HDMNet and our model are 64 and 256, respectively; (3) starting with the same-size image features, e.g., 60×60, HDMNet would downsample them by 4 times (15×15) to calculate attention, while our Mamba scans the features with the original sizes. In brief, we use more blocks, much higher hidden dimensions, and much larger feature sizes, while our model is still much faster than HDMNet. Besides, with the increase of image sizes, the superiority of our method would be much more prominent.

# D  Additional Figures

In this section, the detailed structure of self Mamba block (SMB) is depicted in Section D.1, the visual impacts of query intercepted Mamba (QIM) are highlighted in Section D.2, and more qualitative comparisons between HDMNet and our method on PASCAL-5$^i$ and COCO-20$^i$ are included in Section D.3.

## D.1  Details of Self Mamba Block

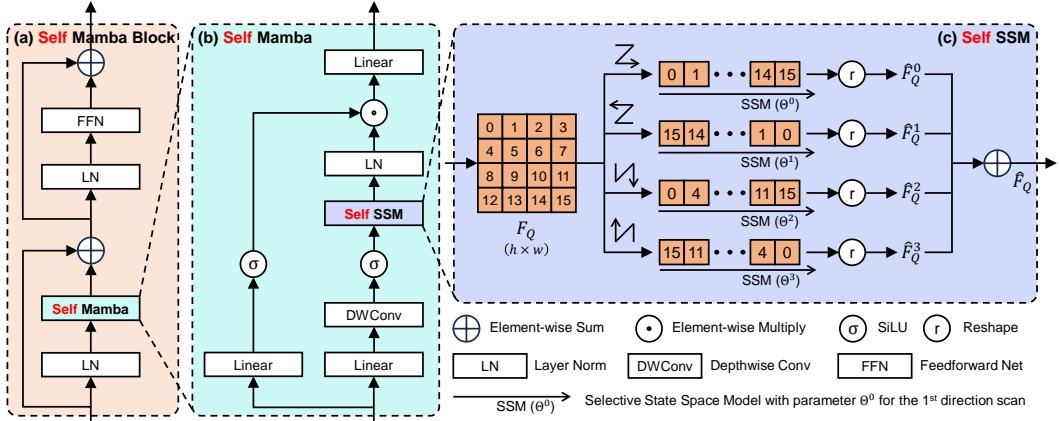

Figure 6: Details of (a) self Mamba block (SMB), devised from an attention block, (b) self Mamba, and (c) self selective selective state model (SSM), taken from VMamba [25].

As illustrated in Figure 6, we implement Mamba blocks in the form of attention blocks [41]. Specifically, a SMB(Figure 6(a)) includes two layer normalization (LN), a self Mamba module, a feedforward network (FFN), and two skip connections. Then, in Figure 6(b), the structure is the same as the standard Mamba [5], while in Figure 6(c), we follow VMamba [25] to use 4 selective state space models (SSMs) to scan the features with 4 different directions, thus, the long-range dependencies can be better captured.

Finally, we would like to emphasize the following points: (1) Each SMB actually contains two branches for separately enhancing query and support features. Particularly, query and support features would share all the layer normalization, as well as the self SSM (Figure 6(c)), for the consistency of feature spaces. Instead, other modules are not shared, because the query and support features are actually a bit different, as the former contains both FG and BG objects, while the latter only consists of FG objects; (2) Hybrid Mamba block (Section 4.2.1) adopts similar structure as SMB, we just need to replace Figure 6(c) as Figure 3.

## D.2  Visualization of Query Intercepted Mamba

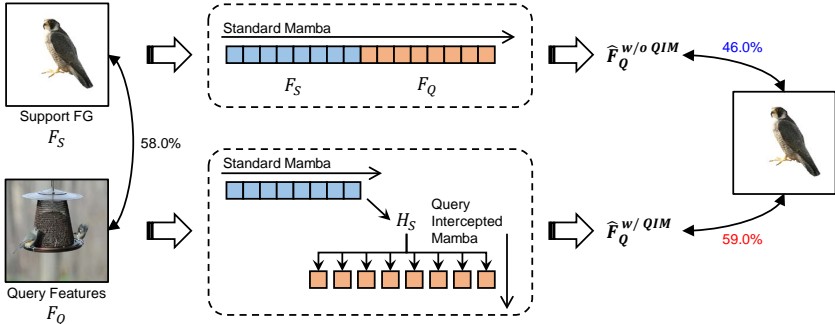

Figure 7: Visualization of QIM module.

QIM is designed to address the *intra-class gap* issue, where query FG objects are essentially more similar to themselves rather than support FG objects, so they may prefer not to fuse information from support FG objects, i.e., the support information is not sufficiently utilized. As shown in Figure 7, we take two models with and without QIM for comparisons. The initial average cosine similarity between query features and support FG features is 58.0%, if we do not use QIM, but adapt existing methods [32, 61] for cross Mamba, the enhanced query features $\hat{F}_Q^{\text{w/o QIM}}$ appears to aggregate more information from themselves, and lead to a similarity drop of 12%, which validates the existence of *intra-class gap* issue. Instead, the similarity between $\hat{F}_Q^{\text{w/ QIM}}$ and $F_S$ can even slightly increase by 1.0%, validating the effectiveness of the proposed QIM.

### D.3 More Qualitative Results

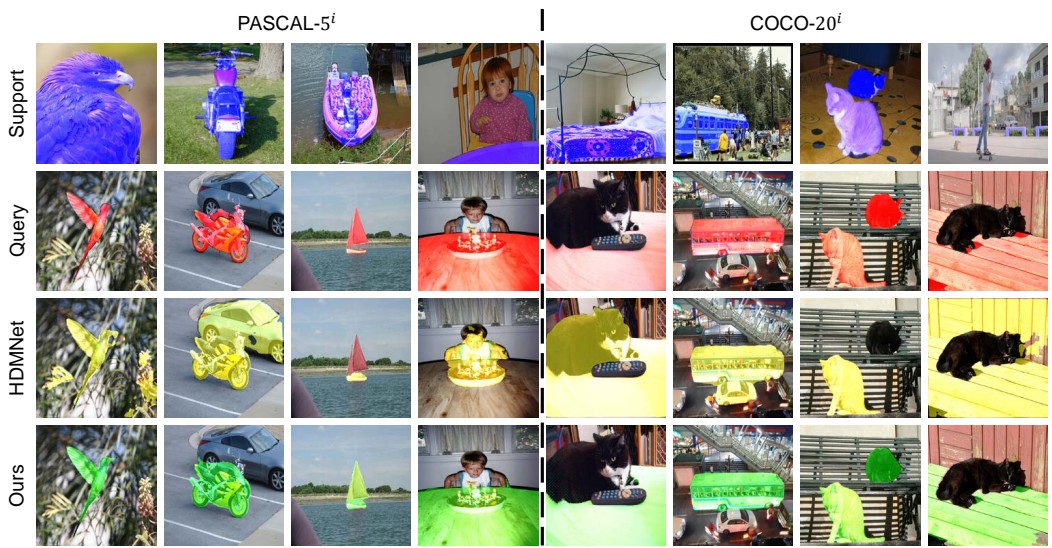

Figure 8: More qualitative results on PASCAL-$5^i$ and COCO-$20^i$.

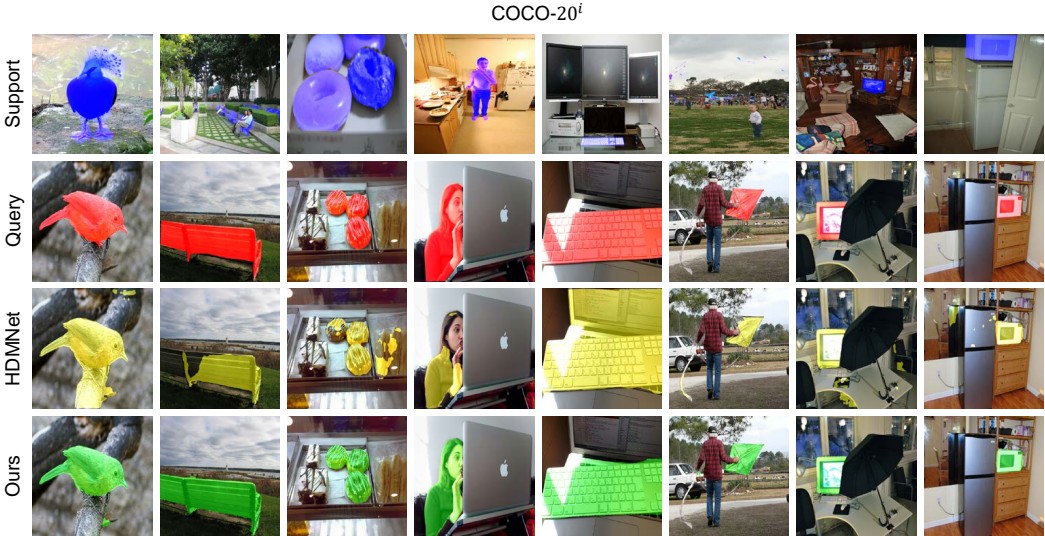

Figure 9: More qualitative results on COCO-$20^i$.

We provide more qualitative results of HDMNet [31] and our model in Figure 8, 9 and 10, where we can witness stable and prominent improvements of our model over HDMNet.

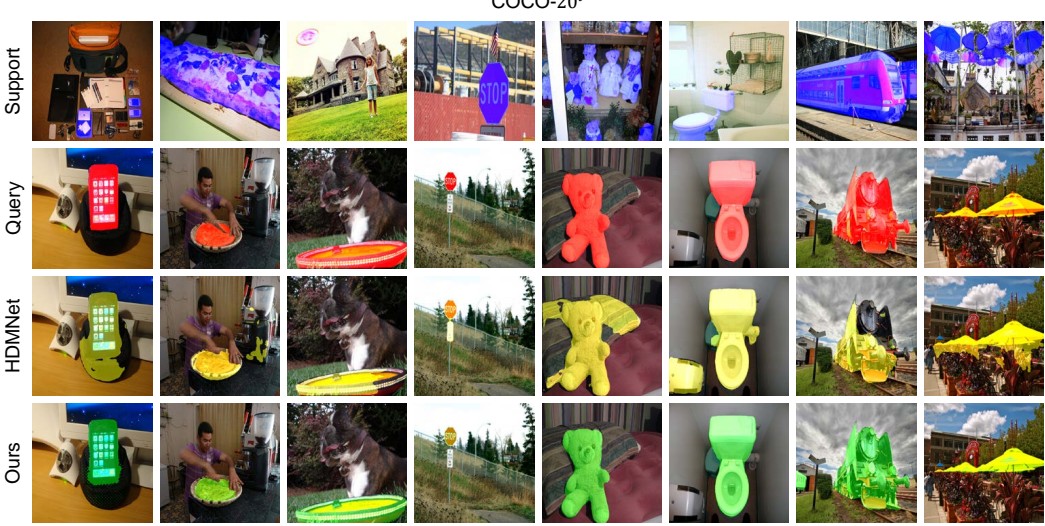

Figure 10: More qualitative results on COCO-20$^i$.

# E    Limitation and Future Direction

To briefly recap, we firstly introduce Mamba into the field of FSS, and present a hybrid Mamba network (HMNet) to effectively fuse query features with support FG features. HMNet further consists of self Mamba block (SMB) and hybrid Mamba block (HMB), where our contributions are mainly in HMB, including a pair of support recapped Mamba (SRM) and query intercepted Mamba (QIM). One limitation of the HMNet is that QIM has not been implemented with CUDA yet, which would make the training and testing time longer. We will include a CUDA implementation in the near future.

Furthermore, Mamba [5] essentially belongs to meta learning, because its input-dependent selection mechanism would project some parameters from the input sequences, and they are used during the scan. Kindly remind that FSS also belongs to meta learning, so a future direction is to explore whether it is feasible to generate query sequence's Mamba parameters directly from the support sequence. A similar work is DPCN [23] which generates dynamic kernels for query features from the support features, and the idea behind is quite different from that of this paper.

