# OpenReview forum: "Hybrid Mamba for Few-Shot Segmentation"
_NeurIPS.cc/2024/Conference — NeurIPS 2024 poster_

### Official Review · Reviewer_noa5 · 2024-07-08

**Soundness:** 3
**Presentation:** 2
**Contribution:** 3
**Rating:** 5
**Confidence:** 4

**Summary:**

This work introduces Mamba to FSS. Particularly, it indicates two issues suffered by the original Mamba when being applied to the cross attention case, namely, the support forgetting issue and the intra-class gap issue.
 To address them, a hybrid Mamba block (HMB) is designed, which can effectively incorporate query FG features with the support
FG features, thereby leading to better FSS performance

**Strengths:**

1. HMB introduces Mamba into FSS, which reduces the computational cost from quadratic complexity to linear complexity.
2. The illustration of HMB is clear and easy to understand.
3. Experiments show the method can achieve state-of-the-art.

**Weaknesses:**

1. Introducing mamba into FFS is to reduce the computational cost, but no experimental results support it. Comparing the efficiency of the proposal method with it of cross-attention based methods is needed.
2. Writing is not standard somewhere. For instance, few-shot segmentation includes few-shot semantic segmentation and few-shot instance segmentation, but the work didn't refer to any instance segmentation task.
3. The content in coco 'train_data_list' file of the supplementary  code  leaks your identity. I don't know if it violates the double-blind rule.

**Questions:**

My questions are written in the above Weakness.

**Limitations:**

This work addresses the limitations(computational cost) in theory， but no experimental results can prove it.
Besides, this work don't have negative societal impact.

---

> ### Author Rebuttal · Authors · 2024-08-06
>
> > W1&L1: Efficiency comparison with cross attention-based methods.
>
> The efficiency comparisons (**different testing episode number**) have been provided in Appendix C.3, where we compare our mamba-based method with a latest cross attention-based method HDMNet [30]. Note that (1) HDMNet employs 6 attention blocks, while we use 8 mamba blocks; (2) the hidden dimensions of HDMNet and our model are 64 and 256, respectively; (3) starting with the same-size image features, e.g., 60x60, HDMNet would downsample them by 4 times (15x15) to calculate attention, while our mamba scans the features with the original sizes. In brief, we use more blocks, much higher hidden dimensions, and much larger feature sizes, but our model is still much faster than HDMNet.
>
> In our response to **Reviewer iSLu**, more efficiency comparisons (**different image sizes**) are provided.
>
> > W2: Standard writing.
>
> Thanks for this comment. We agree it is a bit confused to use 'Few-Shot Segmentation', as our work only focuses on 'Few-Shot Semantic Segmentation'. We would replace the wordings.
>
> > W3: Identity leakage.
>
> We are not the author mentioned in 'train_data_list'. We refer to BAM [15] to build our project, and directly download **their data lists** for experiments, **without any modifications**. Besides, 'train_data_list' is not used in the code, as we directly use their pre-processed lists stored in 'fss_list' folder.
>
> **To summarize, there is no identity leakage problem. The file "train_data_list" is directly downloaded from BAM without modfication.**
>
> Sincerely thanks for your careful checking!

---

### Official Review · Reviewer_iSLu · 2024-07-11

**Soundness:** 3
**Presentation:** 3
**Contribution:** 2
**Rating:** 5
**Confidence:** 2

**Summary:**

This paper proposes a hybrid mamba network (HMNet) to capture inter-sequence dependencies for few-shot segmentation tasks (FSS). The authors identify two issues when applying the original Mamba network to cross-attention scenarios: the support forgetting issue and the intra-class gap issue. The proposed HMNet includes: 1) Support Recapped Mamba (SRM), which periodically recaps the support features when scanning the query to ensure the hidden state contains rich support information; and 2) Query Intercepted Mamba (QIM), which intercepts the mutual interactions among query pixels when propagating the hidden state (with pure support FG) to encourage the fusion of more information from support features. Extensive experiments on PASCAL-5^i and COCO-20^i demonstrate the superiority of the proposed method.

**Strengths:**

1. This paper indicates the support forgetting and intra-class gap issue when applying mamba to FSS.
2. The proposed HMB is more effective for the fusion of support-query features, leading to improved FSS performance.
3. Experimental results on PASCAL-5^i and COCO-20^i verify its effectiveness.

**Weaknesses:**

1. The Mamba model exhibits significantly enhanced efficiency over the attention-based models as the sequence length increases, such as the million-length real-world data mentioned in the Mamba paper. In FSS, is there a comparative analysis that evaluates efficiency and effectiveness across different patch quantity settings?
2. The proposed SRM strategy includes inserting support patches periodically among query patches. Could this approach inadvertently lead to forgetting the query patches in earlier positions?
3. For consistency in terminology, consider distinguishing between "feature" and "patch," as in the caption of Figure 3: "...alternatively appeared support features and query patches...".

**Questions:**

See above.

**Limitations:**

It would be beneficial if the author could address the potential risk of query patch forgetting and include comparisons with attention-based methods across different patch settings.

---

> ### Author Rebuttal · Authors · 2024-08-06
>
> > W1&Q1.2: Comparative analysis for efficiency and effectiveness across different quantity setting?
>
> Thanks for your precious comment. Currently, we compare the efficiency and effectiveness of cross attention-based HDMNet [30] and our method in Appendix C.3, yet only with one image size (COCO dataset - 633x633).
>
> To the best of our knowledge, there is no existing comparative analysis across different image sizes in FSS, because there are no existing benchmark datasets with larger image sizes, so the *effectiveness comparison would be infeasible*. For efficiency, we can interpolate the input query and support images to diverse sizes to **carry out efficiency comparisons**, and we agree it would be better to include such comparisons in our paper. Thanks again for your suggestion.
>
> For this purpose, we interpolate COCO's images to different sizes of {473, 633, 793, 953, 1113} (uniform height and width), and the extracted feature sizes would be of {60, 80, 100, 120, 140}. We compare our mamba method with the cross attention-based HDMNet [30], and the time costs (second) for testing 4,000 episodes (ResNet50, 1-shot, Single 32GB V100 GPU) are as follows.
>
> |Method\Size|473|633|793|953|1113|
> |-|-|-|-|-|-|
> |HDMNet|415.4|676.5|1113.4|1713.7|OOM|
> |Ours|402.4|576.9|768.9|1125.5|1440.6|
> |Diff|13.0|99.6|344.5|588.2|-|
>
> Note that (1) HDMNet employs 6 attention blocks, while we use 8 mamba blocks; (2) the hidden dimensions of HDMNet and our model are 64 and 256, respectively; (3) starting with the same-size image features, e.g., 60x60, HDMNet would downsample them by 4 times (15x15) to calculate attention, while our mamba scans the features with the original sizes. In brief, we use **more blocks**, **much higher hidden dimensions**, and **much larger feature sizes**, while our model is still much faster than HDMNet. Besides, with the increase of image sizes, the superiority of our method would be much more prominent.
>
> > W2&Q.1: Query forgetting issue.
>
> It is true that the interactions among query patches might be weakened. In Figure 3, an idea to avoid query forgetting is to add an extra path, i.e., employ a standard mamba for query features, and fuse the enhanced query features from SRM, QIM and the extra mamba together. However, the mIoU scores of models **with** and **without extra mamba** are 68.2\% and 68.6\%, respectively. Therefore, it's true that query forgetting issue might exist, yet capturing global dependencies for query may not consistently be helpful for FSS.
>
> The success of some cross attention-based methods have validated that non-global attentions can already achieve good performance. For example, our method is similar to SCCAN [52], a swin transformer-based cross attention FSS method, which conducts attention at patch level to capture intra-patch dependencies. Besides, HDMNet [30] downsamples the features by 4 times to perform attention, so the global dependencies are not well captured.
>
> Although capturing global dependencies for query features sounds helpful, the performance might not be improved as expected, verified by our empirical experiments and some existing baselines. One possible reason might be FSS model is trained on some base classes, while directly tested on novel classes, thus, learning to capturing global dependencies for query features during training might learn too much base class-specific information (i.e., inductive bias) that may not be well applied to deal with novel classes.
>
> > W3: Terminology consistency.
>
> Thanks for your suggestion, we would revise the wordings accordingly.

---

### Official Review · Reviewer_hBYs · 2024-07-11

**Soundness:** 3
**Presentation:** 3
**Contribution:** 3
**Rating:** 5
**Confidence:** 5

**Summary:**

This paper proposes HMNet for FSS, which addresses the issues of support forgetting and intra-class  gap in few-shot segmentation by designing support recapped Mamba and query intercepted Mamba, thereby utilizing support information more effectively to enhance segmentation performance. The authors evaluate HMNet through extensive experiments on two public benchmark datasets.

**Strengths:**

1. This paper is the first attempt to apply Mamba to the FSS task.
2. Two mechanisms are designed to adapt the original Mamba to better enhance the query FG features.
3. The proposed method achieves SOTA performance.

**Weaknesses:**

1. The core idea of HMNet appears to be rearranging and combining the support and query features as inputs to Mamba, which may lack a prominent contribution to the design of novel Mamba.
2. To avoid the support forgetting issue, why not use two vanilla Mamba (similar to self-attention) blocks to extract support and query features separately, and then use another Mamba to interact with them (like cross-attention)? Such a mechanism has been evaluated as effective (like HDMNET, CVPR 2023).

**Questions:**

In few-shot segmentation, the background information of support samples may also be helpful in identifying the background regions in the query samples. Have the authors tried to incorporate this aspect into HMNet?

**Limitations:**

The authors have discussed the limitations of the paper.

---

> ### Author Rebuttal · Authors · 2024-08-06
>
> > W1: Core idea of the proposed methodology.
>
> Our main contributions consist of (1) a support recapped mamba (SRM) to address the *support forgetting issue* during the interactions of query and support features, and (2) a query intercepted mamba (QIM) to mitigate the *intra-class gap* issue. Both of them are designed to better fuse query features with support foreground (FG) features for better FSS.
>
> SRM enables the support-query interactions, and we would like to emphasize that (1) we are the first to apply mamba to FSS, the multi-modal data fusion case where the query and support features cannot be pixel-wise aligned; (2) we introduce the naive way to use mamba for support-query interactions, which would suffer from the *support forgetting issue*, i.e., the support information is not well utilized; and (3) we propose SRM to **periodically re-scan the support features**, so as to ensure that the support features (in hidden states) are sufficient enough for enhancing query features. Although SRM is simple, it is the first to facilitate the fusion of two unaligned data sources via mamba, and our experiments can well show its effectiveness.
>
> We also identify the *intra-class gap issue*, and design a QIM to address it. Note that standard mamba is used to **sequentially scan a sequence**, while QIM is designed to **scan each token of a sequence in parallel**, so as to intercept the interactions within query sequence, and encourage query features to fuse more support features, instead of themselves. Such parallel scanning is different from that of the standard mamba, in terms of both rationality and implementation.
>
> > W2: Another attempt (e.g., the form of HDMNet) to address the support forgetting issue.
>
> Thanks for your suggestion! We experiment with the suggested way, i.e., use 4 self mamba blocks to scan query and support features first, then use another 4 hybrid mamba blocks for support-query interaction. However, under the same condition (w/o BAM's ensemble), the mIoU score (PASCAL, 1-shot, ResNet50) of the suggested way is 67.8\%, 0.8\% worse than HMNet. We attribute this to the fact that ***support forgetting issue* has no relationship with self mamba**.
>
> We would like to clarify that the ***support forgetting issue* occurs exactly during the interactions between query and support features (i.e., cross attention/mamba)**. Kindly remind if we would like to fuse support features to query features, a straightforward idea is to scan support sequence first followed by the query sequence, so the support features would be gradually compressed into the hidden states, and be fused into query features when scanning query features. However, the problems are (1) both support and query sequences are long, e.g., more than 3,600 feature vectors/tokens in FSS; (2) with the scan on query, query features would also be gradually compressed into the hidden states, i.e., *the proporation of support information (in hidden states) would gradually decrease, so many query vectors may not be able to fuse sufficient support information, especially for those vectors at end*. Therefore, we design the support recapped mamba to address this issue.
>
> > Q1: The use of support background (BG).
>
> Thanks for this comment. Most of existing FSS methods only consider support FG information, because (1) for the same FG class, the BG objects contained in query and support images are usually different, e.g., human (query/support FG) can sit on a sofa (query BG), or play in the wild (support BG), where the support BG is different from both query FG and BG, so it may not be useful; and (2) the use of support BG might lead to the *overfitting issue*, as the BG classes of base FG classes (for training) and novel FG classes (for testing) can be quite different. Therefore, the model can easily get overfitting to deal with the *base class-specific BG features*, which cannot be well applied to process the **different BG features of novel classes**.
>
> There are few special designs made to utilize support BG information, but the method would be quite different to the current ones, and we think it's a good idea to leave it as a future direction.

---

### Official Review · Reviewer_dFqt · 2024-07-18

**Soundness:** 2
**Presentation:** 3
**Contribution:** 2
**Rating:** 4
**Confidence:** 5

**Summary:**

This work proposes a hybrid Mamba network for few-shot segmentation, including a support recapped Mamba to periodically recap the support features when scanning query and a query intercepted Mamba to forbid the mutual interactions among query pixels.

**Strengths:**

The idea of adapting Mamba for few-shot segmentation is good. The performance looks good.

**Weaknesses:**

1. The motivation of using mamba for FSS is unconvincing. The authors state that they use mamba for FSS is because that mamba's complexity is only linear. However, the proposed mamba-based method is used only for fuse support foreground (FG) into query features. There are also many light-weight feature fusing methods for FSS and show good performance and efficiency. The authors need to provide convincing reasons how the mamba works better than other feature fusing methods. Otherwise, just applying a new technology to FSS without deep insight is not acceptable.
2. The experimental analysis is insufficient. First, the comparison experiments between the proposed mamba methods and other feature fusing methods are missing. I cannot know if the proposed method is better, on performance or efficiency. I mean the fair comparison on a same baseline method to compare with other methods, rather than just the Table 1 and Table 2. The overall performance comparison is unfair because the baseline methods are different. Second, the analysis is insufficient. The authors state that the proposed mamba methods solves the support forgetting and intra-class gap issue problems. But how these problems are solved? The Table 3 and Figure 5 just show that the proposed method achieves better performance. However, it is still unclear how the proposed method solve these problems.
3. It also lacks the comparison with SAM-based FSS methods.

**Questions:**

See weakness

**Limitations:**

No.

---

> ### Author Rebuttal · Authors · 2024-08-06
>
> > W1: Motivation of using mamba for FSS.
>
> Thanks for this comment, and we agree it's better to make the motivation stronger:
> 1. (Discussion) Kindly remind FSS is a task where the model is trained on some base classes, and directly applied to test novel classes. During training, the model can easily *get biased*, i.e., the learned parameters *overfit base classes*, and are *inappropriate for novel classes*. Note that the main idea of mamba is the **selection mechanism**, making **mamba parameters input-dependent**. Hence, mamba might mitigate the issue well, as its parameters can be **dynamically generated based on the test data**.
> 2. There are many works showing that self mamba can well surpass self attention in both **efficiency** and **effectiveness**, e.g., [24, 63] validate this point in various vision tasks such as image classification, object detection, and **semantic segmentation**. Therefore, it's reasonable to apply mamba to **few-shot semantic segmentation**.
> 3. As there are two data sources (query and support), FSS models require more GPU memory, e.g., recent advances like HDMNet [30] uses self attentions to *separately* process query and support features. Given that attention has *quadratic complexity*, while mamba only has **linear complexity**, mamba is suitable for FSS to reduce the computational burden. In our response to **Reviewer iSLu**, we have provided some clues showing that our mamba is much **faster** than HDMNet, and can deal with **larger images**.
> 4. The key to the success of FSS lies in the better use of support pairs, and recent methods usually use cross attention to fuse support FG into query features. Unfortunately, there is *no existing cross mamba* for this case, so **it's necessary to introduce a cross mamba to enable efficient support-query interactions**, which is our main motivation.
> 5. However, it's challenging to achieve such goal. In Section 1, we introduce a naive way, but it would suffer from the *support forgetting* and *intra-class gap issues*. Accordingly, we design **SRM** and **QIM** to address them.
>
> > W2.1: Compare with feature fusing methods.
>
> Such comparison is provided in L256-L258, where we fairly compare our method with SCCAN [52], but we do agree it's better to include more methods. Hence, we further select 1 feature concatenation method [15] and 2 attention methods [52, 58] for comparisons.
>
> |Method|$5^0$|$5^1$|$5^2$|$5^3$|mIoU|
> |-|-|-|-|-|-|
> |Concatenation [15]|65.6|71.5|64.6|59.2|65.2|
> |Cross Attention 1 [58]|67.0|71.5|65.9|57.1|65.4|
> |Cross Attention 2 [52]|68.3|72.5|66.8|59.8|66.8|
> |Ours|70.1|73.6|67.7|62.9|68.6|
>
> Note that the values of [15, 58, 52] are reproduced on the same codebase as us. Specifically, (1) we remove the unfair ensembles of BAM [15], and re-train the **feature concatenation** model; (2) CyCTR [58] designs the **cycle-consistent attention (cross attention 1)**, and we fairly reproduce its results; (3) the comparisons with SCCAN [52] is the fairest, which designs a **self-calibrated cross attention (cross attention 2)**. Particularly, we both use 8 attention/mamba blocks, uniform prior masks, hidden dimensions, etc.
>
> From the table, we can observe (1) our model behaves much better than others; (2) in the fairest case, our cross mamba is more effective than cross attention [52], showing the superiority of our method.
>
> > W2.2: How the issues are resolved.
>
> The naive cross mamba is to put support in front of query features, and use a standard mamba to scan them. After scanning support, the hidden states (of mamba) would contain the summary of support FG features, which would be fused into query features when scanning on them.
>
> Unfortunately, there exist two issues:
> - *Support forgetting*: The support FG features in hidden states would be gradually replaced by query features, e.g., query BG would also be compressed into the finite hidden states. Hence, many query feature vectors (especially those at end) cannot fuse sufficient support FG features from the hidden states;
> - *Intra-class gap*: Although query and support FG belong to the same class, their features can be quite different, e.g., two people are visually different. When scanning on query, the hidden states include both support and query FG features, so query FG (people 1) may prefer to fuse more compressed query FG (people 1), instead of support FG features (people 2).
>
> To address them, we design:
> - **SRM**: We split query features into patches, and before scanning each of them, we re-scan support features to ensure there exist sufficient support information in the hidden states;
> - **QIM**: We propose to forbid compressing query features into the hidden state, i.e., after scanning support, the same hidden states would be used to scan each query feature. In this way, query FG (people 1) cannot find compressed query features (people 1), but to fuse support FG (people 2).
>
> To validate their effectiveness, we conduct both quantitative (Table 3) and qualitative analysis (Figure 5 and 7):
> - Figure 5 and Table 3 validate (1) the existence of *support forgetting*, as the similarity between support FG and the hidden states decreases after scanning each query patch; (2) the effectivess of **SRM**, as the similarity can be recovered back to high values after re-scanning; (3) the improvement is good.
> - Figure 7 and Table 3 validate (1) the existence of *intra-class gap*, as the similarity between support FG and query features (after mamba) is only 46\%, showing the insufficient fusion from support to query; (2) the effectiveness of **QIM**, as the similarity can be improved to 59\%; (3) the improvement is good.
>
> To summarize, both **SRM** and **QIM** are proposed to **facilitate fusing more support FG features into query features**, leading to more effective FSS.
>
> > W3: Compare with SAM-based methods.
>
> Both with BAM's ensembles, our model behaves better than FSS-SAM, and the results are included in the **uploaded PDF**.

---

### Author Rebuttal · Authors · 2024-08-06

Dear Reviewers, ACs, SACs, and PCs,

Thanks for your insightful feedback on our manuscript: Hybrid Mamba for Few-Shot Segmentation!

We are encouraged that you find our contribution is good (hBYs, noa5), the soundness is good (hBYs, iSLu, noa5), the motivation of adapting Mamba for FSS is reasonable (all reviewers), the methodology is effective (hBYs, iSLu), the paper is well written (all reviewers), and the performance is good (all reviewers).

Within this short rebuttal period, we did our best to reply to your questions, and we will incorporate your feedbacks in the final version. In the upcoming interaction period, if you have any questions towards our responses, please feel free to let us know.

Kindly remind that we attach a **PDF**, containing the tables to compare our method with a SAM-based FSS method, as required by **Reviewer dFqt**.

Sincerely,

Authors

---

### Author Response · Authors · 2024-08-11
**Request for Discussions**

Dear Reviewers, ACs, SACs and PCs,

Thanks again for your detailed review and constructive suggestions for improvement. We have carefully considered your comments and suggestions, and provided detailed responses.

As the discussion period is concluding in 3 days, we kindly request you to read our rebuttal at your earliest convenience. We are hopeful that our responses adequately address your concerns, and you can reconsider your score.

If you require further clarification on any points, please do not hesitate to reach out. We are willing to provide additional information to ensure your confidence in the responses made.

Sincerely,

Authors

---

### Decision · Program_Chairs · 2024-09-25

**Decision:**

Accept (poster)

**Comment:**

One review asked a good question about whether using Mamba is indeed necessary if it is mainly doing smart feature aggregation.  The authors did not seem to answer this in their rebuttals.  Overall the reviewers found that despite issues in clarity that the submission offered a novel contribution and enough experimental evidence to accept.  Please take review comments into account to make the paper clearer.